# Female genital mutilation/cutting: Emerging factors sustaining medicalization related changes in selected Kenyan communities

Samuel Kimani[1,2]*, Caroline W. Kabiru[3], Jacinta Muteshi[3], Jaldesa Guyo[2]

1 School of Nursing Sciences, University of Nairobi, Nairobi, Kenya, 2 Africa Coordinating Centre for the Abandonment of Female Genital Mutilation /Cutting (ACCAF), Department of Obstetrics and Gynecology, University of Nairobi, Nairobi, Kenya, 3 Population Council-Kenya, Nairobi, Kenya

☺ These authors contributed equally to this work.
* tkimani@uonbi.ac.ke, thuo.kimani@gmail.com

**Data Availability Statement:** The data underlying this study have been uploaded to OSF and are accessible using the following link: https://osf.io/

## Abstract

Although female genital mutilation/cutting (FGM/C) has declined, it is pervasive albeit changing form among communities in Kenya. Transformation of FGM/C include medicalization although poorly understood has increased undermining abandonment efforts for the practice. We sought to understand drivers of medicalization in FGM/C among selected Kenyan communities. A qualitative study involving participants from Abagusii, Somali and Kuria communities and key informants with health care providers from four Kenyan counties was conducted. Data were collected using in-depth interviews (n = 54), key informant interviews (n = 56) and 45 focus group discussions. Data were transcribed and analyzed thematically using NVivo version 12. We found families practiced FGM/C for reasons including conformity to culture/tradition, religion, marriageability, fear of negative sanctions, and rite of passage. Medicalized FGM/C was only reported by participants from the Abagusii and Somali communities. Few Kuria participants shared that medicalized FGM/C was against their culture and would attract sanctions. Medicalized FGM/C was perceived to have few health complications, shorter healing, and enables families to hide from law. To avoid arrest or sanctions, medicalized FGM/C was performed at home/private clinics. Desire to mitigate health complications and income were cited as reasons for health providers performing of FGM/C. Medicalization was believed to perpetuate the practice as it was perceived as modernized FGM/C. FGM/C remains pervasive in the studied Kenyan communities albeit changed form and context. Findings suggest medicalization sustain FGM/C by allowing families and health providers to conform to social norms underpinning FGM/C while addressing risks of FGM/C complications and legal prohibitions. This underscores the need for more nuanced approaches targeting health providers, families and communities to promote abandonment of FGM/C while addressing medicalization.

z7rm8 and the DOI: https://doi.org/10.3886/
E117701V1.

**Funding:** The study was funded by the UK Aid from the UK government under the DFID research project "Evidence to End FGM/C: Research to Help Women Thrive," which is coordinated by the Population Council. All content is the sole responsibility of the authors and does not represent the opinions of the UK Government or the Population Council. The funders had no role in study design, data collection and analysis, decision to publish, or preparation of the manuscript.

**Competing interests:** The authors declare that they have no conflict of interest regarding the publication of this paper.

## Introduction

The practice of female genital mutilation/Cutting (FGM/C) is considered a form of abuse, undermines girls' and women's dignity and violates their human rights. [1] FGM/C includes all procedures involving partial or total removal of the female external genitalia or other injuries for non-medical reasons. [2] Although the World Health Organization (WHO) has developed a clear typology of FGM/C to include: clitoridectomy-type I, excision-type II, infibulation-type III and all other harmful procedures to the female genitalia for non-medical purposes (type IV) [1], the practice has undergone considerable changes (shifts) in form and context. Some of the changes involve communities that traditionally performed severe forms (infibulation) but currently practice less severe (type II, type I or IV) cutting. Additionally, even though FGM/C performed by health care providers is considered less severe, there is poor understanding of the category as per the WHO typology. Although the performance of FGM/C by health providers is not entirely a new phenomenon, it is of great interest because girls are increasingly undergoing the procedure as well as it presents a new challenge in the abandonment efforts.

Globally, an estimated 200 million women/girls have undergone FGM/C, while a further 3.6 million girls risk being cut annually. [3] The highest prevalence of FGM/C has been recorded in 28 African countries, Asia, the Middle East, Latin America and among migrant populations from practicing communities in some Western nations. [4] In Kenya, the prevalence among women and girls aged 15–49 years was estimated at 21 percent in 2014—a decline from 38 percent in 1998 with notable variation across ethnicity, regions and age cohorts. [6] Except for the Luo, Luhya, Pokomo, Teso and Turkana, all other Kenyan ethnic groups have practiced FGM/C at one point or another. [5, 6] The prevalence is affected by the quality and ability of FGM/C module included in the house hold surveys to address bias in the light of some communities not considering medicalization and less severe cutting as a form of FGM/C.

The persistence of FGM/C is attributed to time and generational linked social norms observed and adhered across different practicing communities. These norms have sustained the practice albeit changes in form and context. Noticeable shifts in FGM/C include cutting by health professionals (medicalization), [7, 8] less severe cutting, and cutting girls at a younger age than previously done. [9, 10] Medicalization represent a change that entails situations where health care professionals (doctor, nurse, midwife, or other health professionals) perform FGM/C either in a health facility, at home or a neutral place, often using surgical tools, anesthetics and antiseptics. [7, 8] It also includes re-infibulation—re-closing external genitalia of women who had been de-infibulated to allow for sexual intercourse, birth delivery, and/or related gynecologic procedures by doctors or nurse-midwives. [7, 11] Although these shifts are supported by community social norms passed for generations, other dynamics appear to play important role. Emerging evidence from demographic and health surveys (DHS), multiple indicator cluster surveys (MICS), and qualitative research suggests that some families and communities are shifting how FGM/C is practiced, to sustain rather than abandon it, mainly due to–reduce the health risks, willingness of some health providers to carry out the procedure, financial incentive or social recognition. [6, 7, 9] The greatest burden of medicalized FGM/C is concentrated in Sudan (67%), Egypt (38%), Guinea (15%), Kenya (15%) and Nigeria (13%) where nurses, trained midwives, or other lower-level providers perform FGM/C, Egypt is exceptional because cutting is mainly by doctors [6, 9] Furthermore, the risk of medicalization is higher among girls aged 0–14 years than among women aged 15–49 years and a trend towards medicalization is more likely to institutionalize the practice and encourage its continuation rather than abandonment. [7, 12, 13]

The reasons and dynamics for the perpetuation of FGM/C and related shifts among different ethnic groups in Kenya are poorly understood. In this manuscript, we focus on the medicalization of FGM/C in three selected communities (Abagusii, Somali and Kuria) in Kenya where the practice remains pervasive. A comprehensive narrative regarding FGM/C practice in the three communities has been provided in the methods section. A mother-daughter comparisons using data from Kenya Demographic and Health Survey 2014, show an increase in medicalization [6, 9] with about 20 percent of girls younger than 15 years having been cut by health professionals compared to 15 percent of those aged 15–49 years. [5, 6] Ethnic groups with substantially high medicalization rates in Kenya include the Abagusii [10, 14, 15]; Somali [10, 16]; and the Maasai. [17] Medicalization has also been linked with other changes in FGM/C, notably less severe cutting—in the form of nicking, pricking and scraping (Type IV FGM/C)—and cutting at a younger age, both of which have been increasing. [9, 10] Whether these changes have direct relationship to medicalized FGM/C is subject for investigation.

Evidence show that medicalization of FGM/C is adopted for reasons that span from the community as well as the health providers to include: reducing the risk of immediate complications, conforming to social norms, financial gains and social recognition for the health practitioners. [11, 14, 18, 19] Furthermore, health care providers from practicing cultures believe medicalization is acceptable, promotes quick recovery and could help evade law enforcement. [9, 10] Medicalization has however, been condemned and challenged because it does not prevent long-term medical, psychological or sexual complications associated with the practice. [10] On the contrary, medicalization is believed to legitimize and encourage the continuation of FGM/C among practicing communities as health care providers are respected members of society likely to be emulated. [7, 8] Although policies and strategies for addressing the abandonment of FGM/C have been developed, fewer strategies exist to tackle increasing medicalization, while their effectiveness have not been ascertained possibly because of paucity of data. In this study, we therefore sought to understand the drivers of medicalization in FGM/C in selected Kenyan communities.

## Materials and methods

### Study design

The data are from a larger cross-sectional qualitative study that sought to understand shifts in the practice of FGM/C in selected practicing communities in Kenya. [10] The study strived to understand why families sought FGM/C for their daughters from health care providers and why the providers accepted to cut the girls. The general context of FGM/C in three (Somali, Abagusii, and Kuria) Kenyan communities were elicited. Specifically, changes in FGM/C including cutting girls less severely and at younger age as well as medicalization of the practice were explored in participants from the Somali, and Abagusii communities who resided in urban and rural settings, as well as among the Kuria in their rural homes. Data were collected between December 2016 and November 2017 using key informant interviews (KIIs), in-depth interviews (IDIs) and focus group discussions (FGDs).

### Study location

The study was conducted in four counties (Garissa, Kisii, Migori, and Nairobi). In Garissa, the study was conducted in Garissa Township sub county, which has a large Somali population. The study was conducted in Gucha and Gucha South sub-counties located in the former Nyanza province to target the Abagusii community in Kisii County. Data were collected in Kuria East and Kuria West constituencies which is inhabited by the Kuria community in

Migori County. In the city of Nairobi, data were collected in Eastleigh a place with high population of Somalis while Kawangware has a high proportion of the Abagusii.

## Study participants

The participants included members from the Abagusii, Kuria and Somali communities. The Abagusii, comprise the sixth largest ethnic group. [5, 15] The prevalence of FGM/C among the Abagusii is high of 86 percent as of 2014. [5] The community traditionally practiced FGM/C as a rite of passage (above 10 years) to adulthood as well as because of strong generational cultural and traditional reasons. Those who are cut have high prospects of being marriage, are respected, considered mature and their spouses held in high status. Those who are not cut are negatively sanctioned, considered immature, despised and ridiculed and could be cut when mature especially while giving birth. The mothers in law and grandmothers are very influential in determining who and when FGM/C would be performed. There is strong evidence that FGM/C is performed on younger ages (less than 10 years) as well as medicalization in Abagusii [14]. The FGM/C is performed discreetly at home involving one or fewer girls performed by health care providers including the retired ones. The health care providers make financial income by performing FGM/C. On the other hand, the Kuria community belong to Bantu ethnic group also found in Tanzania. The practice of FGM/C among the Kuria is almost universal with a prevalence of 96 percent as of 2014. [5] The community practice circumcision as a rite of passage (above 12 years) to adulthood, as a culture and tradition for both girls and boys. The community hold public FGM/C ceremonies on dates decided by the council of elders. [20] The council of elders are the most influential custodians of the practice, dictate when, who, and where the FGM/C would be performed as well as the practitioners to be involved. The elders benefit financially and materially from the FGM/C practice. The Somalis on the other hand, are the largest Cushitic ethnic group in Kenya and predominantly Muslims. There is evidence of the practice being performed on girls younger than 10 years old than it previously used be. Historically, Somali women underwent type III FGM/C—also referred to as Pharaonic circumcision, which is traditionally seen as proof of virginity when young women get married. [10, 21, 22] However, recently a shift towards less severe cutting, often referred to as "Sunna" circumcision, has been documented. [6, 10, 16, 21] Somalis practice FGM/C to conform with culture/traditions, enhance girls' marriageability, for aesthetics, virginity and purity, to avoid social sanctions, as well as for religious reasons. [23–26] The girls are cut in homes or private clinics while no public cerebrations are conducted.

The participants included cut women of reproductive age (15–49 years) from the three communities as well as mothers of girls who had been cut (medically or traditional). Husbands or male partners of the cut women or girls were also included. We also interviewed health care providers (doctors, nurses, midwives and clinical officers), traditional birth attendants, community leaders (diverse professionals, administrative officials, religious leaders and officials from community organizations), and representatives from local non-governmental organizations' (NGOs).

## Recruitment, consent process and interview procedures

Participants were recruited using purposive and snowball sampling. Locally networked community-based organizations and local administrators helped recruit the initial study participants. Thereafter, snow-ball sampling was used to identify additional participants through referrals from the initial group of participants. Once participants were identified, the researcher approached the household head or the facility administrator, explained the purpose of the research, and obtained permission to conduct the study. Once identified and permission

granted, the participants were taken through the consenting process to understand: study components, risks involved, confidentiality, any compensation, and the freedom to withdraw from the interview without suffering or being punished. Thereafter, the participants were requested to sign or give a verbal consent to participate as well as allow for audio recording.

Study participants aged 18 years and older granted informed consent, while assent was obtained from younger (minors) participants, in which explanation on the study was done but the consent was provided by the parents or legal guardian for participation into the study and for audio recording of the discussion/interviews. The discussions/interviews were conducted in private spaces and at times convenient to the participants. The interviews/discussions lasted for about one and a half hours and were conducted in Somali, Abagusii, Kuria, Swahili or English by trained locally-recruited research assistants. The discussion/interview guides contained specific questions that addressed the highlighted issues of medicalization and related changes, but also there were probes to identify specific issues. Interviews/discussions were audio recorded, with permission from the participants, for data retrieval and accuracy. Separate FGDs were conducted with younger, middle aged, and older women and men. Each FGD had 6–12 participants. The final sample size (54 IDIs, 56 KIIs and 45 FGDs) was informed by thematic saturation of data as summarized in **Table 1**. Data from the different categories of participants were triangulated to generate a comprehensive list of issues regarding FGM/C including its medicalization. Ethical approval for the study was granted by the Population Council's Institutional Review Board (Ref: 775; dated: November 9, 2016), and the Kenyatta National Hospital-University of Nairobi (KNH-UoN; Ref: P527/07/2016; dated: October 12, 2016). Permission to carry out the study was granted by the National Commission for Science, Technology and Innovation (Ref: NACOSTI/P/16/79790/14328; dated: October 31, 2016) and county administrators. Study participants aged 18 years and older granted informed consent. For participants younger than 18 years, we obtained both parental consent and child assent.

## Data management and analysis

Demographic characteristics data namely gender, age, ethnicity, marital status, location as well as cadre, work station and organization represented for the participants were entered in anonymized form into password-protected Excel spreadsheets and descriptively analyzed. Digital audio recordings of the group discussions and interviews were subjected to a multi-stage translation-transcription process to ensure data quality. First, recordings in local languages and Kiswahili were transcribed verbatim by experienced local language-or Kiswahili-speaking transcribers. Second, the anonymized transcripts were independently reviewed by target language-speaking translators, who checked the transcripts against the original audio recordings for accuracy, spelling and content. Any differences detected between the two formats were identified, discussed and resolved between the original transcription and the translator. Third, finalized transcripts were translated from local languages and Kiswahili to English by bilingual translators. A sample (10%) (n = 15) of the transcripts were reviewed by three independent reviewers and differences between the original English translation and the reviewed samples were identified and anomalies discussed until consensus on accurate translation was achieved. The finalized translated versions were then subjected to qualitative analyses. The transcripts were de-identified and stored in a password-protected computer.

The framework method for qualitative content analyses [27] was adopted for this study. The method is appropriate for thematic analysis of textual data where it is important to compare data by themes across many cases. [28] This approach combines deductive and inductive analyses of textual data with the flexibility to adapt emerging data and produce a coding framework, or 'template'. The themes/codes were selected through a combined approach:

**Table 1. Demographic characteristics of study participants by interview type.**

| Characteristic | Communities | | | | | | | | | Total |
|---|---|---|---|---|---|---|---|---|---|---|
| | Somali | | | Kisii | | | Kuria | | | |
| | FGD | IDI | KII | FGD | IDI | KII | FGD | IDI | KII | |
| **Gender** | | | | | | | | | | |
| Male | 8 | 1 | 14 | 4 | 6 | 8 | 1 | 3 | 9 | 54 |
| Female | 12 | 3 | 18 | 12 | 26 | 10 | 8 | 7 | 5 | 101 |
| **Age (years)** | | | | | | | | | | |
| Not indicated | 11 | 4 | 28 | 6 | 2 | 4 | 7 | 1 | 2 | 65 |
| below 14 | 5 | | 2 | 3 | 1 | | | | | 11 |
| 14–17 | | | | 2 | 11 | | 2 | 2 | | 17 |
| 18–49 | | | 1 | 2 | 13 | 10 | | 2 | 8 | 36 |
| Above 49 | 4 | | 1 | 3 | 5 | 4 | | 5 | 4 | 26 |
| **Education** | | | | | | | | | | |
| Not indicated | 19 | 4 | 17 | 16 | 26 | 11 | 9 | 8 | 3 | 113 |
| Primary | | | | | | | | 2 | | 2 |
| Secondary | | | | | 3 | | | | | 3 |
| Tertiary | 1 | | 14 | | 2 | 7 | | | 11 | 35 |
| **Marital status** | | | | | | | | | | |
| Not indicated | 18 | 3 | 28 | 15 | 1 | 5 | 9 | 1 | | 80 |
| Not married | | | 1 | 1 | 12 | 3 | | 2 | 1 | 20 |
| Married | 2 | 2 | 2 | | 19 | 10 | | 7 | 13 | 55 |
| **Religion** | | | | | | | | | | |
| Not indicated | | | 4 | | | | | | | 4 |
| Islam | 20 | 4 | 27 | | | | | | | 51 |
| Christians | | | | 16 | 32 | 18 | 9 | 10 | | 85 |
| **Profession** | | | | | | | | | | |
| Not indicated | 16 | 4 | 8 | 16 | 18 | 7 | 9 | 10 | 6 | 94 |
| House wife | | 1 | | | | | | | | 1 |
| Healthcare Providers | 2 | | 14 | | | 4 | | | 3 | 23 |
| Chief | | | 3 | | | 2 | | | 1 | 6 |
| Business | | | 2 | | 8 | | | | | 10 |
| Religious Leader | 1 | | 3 | | | 2 | | | 2 | 8 |
| Students | 1 | | | | 6 | | | | | 7 |
| Politician | | | 1 | | | | | | | 1 |
| Teacher | | | | | | 3 | | | 2 | 5 |

deductively based on previous literature and the specifics of the research question, as well as inductively from the transcripts, followed by their refinement. Further details in the development of the coding framework and the quality assurance processes are described in the main study report. [10]

## Results and discussion

The findings are presented and structured around the following themes: reasons families practice FGM/C including medicalization as well as why health care providers accept to perform FGM/C. The findings are compared across the three Kenyan ethnic groups. Themes and sub-themes from the data are summarized in **Fig 1**.

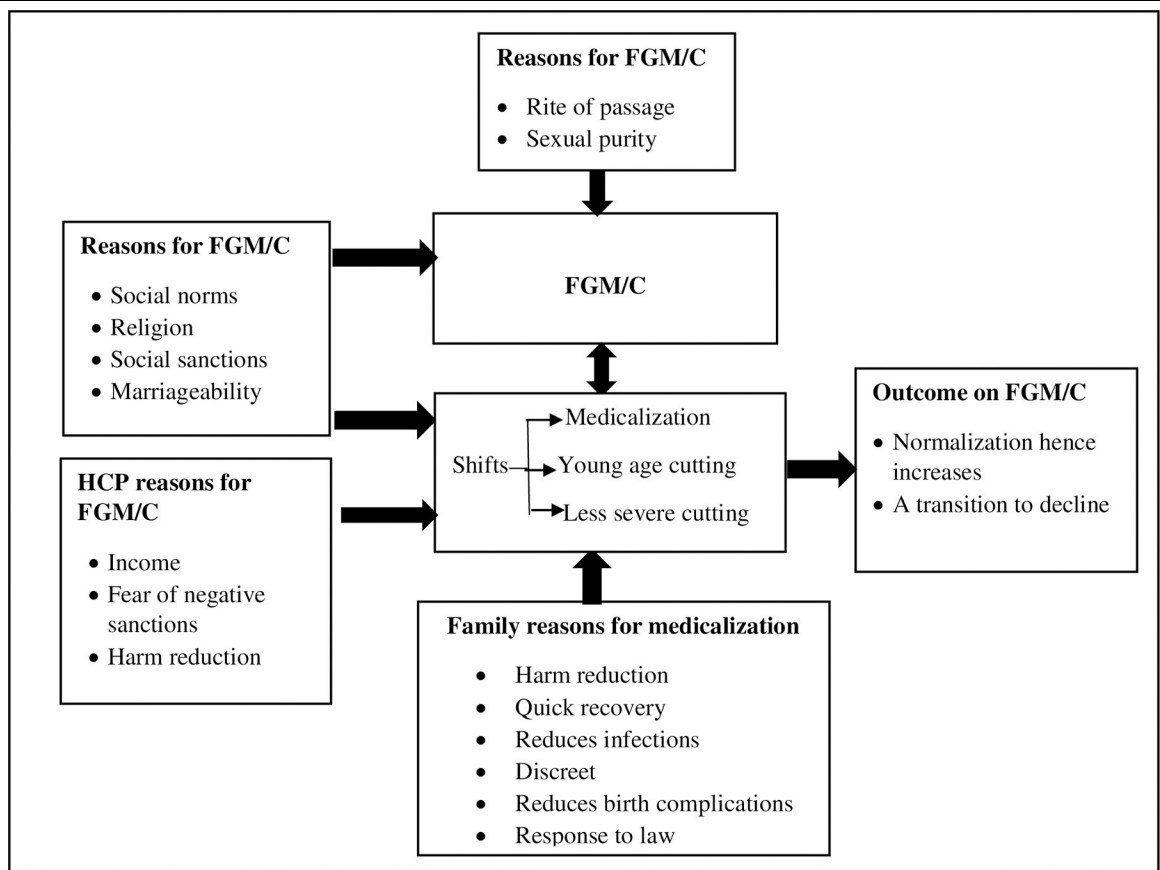

**Fig 1. Summary of reasons that perpetuate FGM/C including medicalization in three Kenyan communities.** Various reasons perpetuate FGM/C including medicalization among families and health care providers in the Kenyan Communities. Although traditional factors still sustained FGM/C and the related shifts, contemporary practice of FGM/C is perpetuated by new sets of reasons.

### Reasons why families practice FGM/C

The persistence of FGM/C in the three communities was associated with social norms that have been observed for generations. Participants from the three communities noted that FGM/C was a cultural and traditional practice that members were obliged to perform as a form of respect to forefathers. For the Abagusii, FGM/C was noted to distinguish them from non-circumcising, neighboring ethnic groups like the Luhya and Luo thus offering a sense of identity. To some, conformity to cultural practices was critical. As a member of the council of elders, an entity trusted with the custody of culture in the Kuria community, emphasized with an angry tone, *"FGM/C is a culture and tradition we have to respect. It was a tradition from our forefathers meant for both boys and girls. It is very bad when you talk of abandoning our culture."*

Religion as a basis for FGM/C was only highlighted by Somali participants, who are predominantly Muslims. Although there were varied views on whether FGM/C is obligatory under Islam, many participants noted that *Sunna* cutting—which was often described as a milder form with few complications—is supported by religion as described in the hadiths of Prophet Mohamed. As noted by an older man who participated in an FGD in Eastleigh, *"If we follow the hadith, the top most part of the clitoris is to be cut to make blood come out, this has no problem. It is the Sunna of the prophet (Peace Be Upon Him)."* FGM/C was therefore perceived by some as a rite of purification that enabled girls to participate in religious prayers. A few

participants, however, noted that all types of FGM/C are prohibited in Islam and that no part of the Quran supports it. According to them, the lack of a religious basis for FGM/C had resulted in the abandonment of the practice by some people in the community.

Participants from Kuria noted that FGM/C is a rite of passage that marked transition from childhood to adulthood. As one grandmother who participated in an interview indicated *"Circumcision has been there, and every woman has to pass through it before being married and for her to be called an adult."* Participants explained that undergoing FGM/C also gives girls a sense of belonging with peers who undergo the rite at the same time while those who do not are isolated during social functions.

Across the three communities, FGM/C is regarded as an important prerequisite for marriage. Women and girls who are cut are considered mature, respectful, and faithful in marriage. Cut women and girls are therefore thought to have better marriage prospects, as illustrated by the words of a married Abagusii man living in Kawangware who quoted, *"The Abagusii people believe that when a girl is cut, that's when she can find a husband. . .In our culture someone could not marry a girl who is not cut, it will mean that you have married a child. Yes, some say they won't marry uncut girls, commonly known as 'egesagane', (laughs) meaning she does not have respect."* Similar sentiments were shared by participants from other communities. Although the interviews suggested that some women did get married without undergoing FGM/C, some participants from the Abagusii community shared that such women would be cut during childbirth by either a traditional cutter or health care providers.

Closely related to the practice of FGM/C enhancing marriageability, there was a belief that FGM/C ensures girls' sexual purity. Across the three communities, responses suggest that it is widely believed that cutting the clitoris lowers girls' libido and prevents premarital sex, immorality or prostitution. An NGO worker in Kuria, for example, noted, *"I have interacted with the community, and the various groups, that is, the council of elders, the mothers, men and the young people. FGM/C is still seen to reduces the sexual urge of the girls."* Sexual purity was often highlighted as an important prerequisite for marriage that would confer dignity on the family. Participants from the Somali community noted that infibulation, in particular, marked virginity and purity. As such, some participants claimed that uncircumcised girls lacked suitors or faced divorced upon the discovery that they have not been infibulated.

Given the value placed on FGM/C, participants from the three communities noted that girls face intense social pressure to undergo the practice. Claims that girls and women who are uncut are discriminated against and stigmatized were commonplace. For example, uncut girls were reportedly called derogatory names, disrespected and shamed. Families, too, faced social pressure to have their daughters cut even when they were not in supportive of the practice. Explaining how parents were "forced" to cut their daughter, a married woman in Kuria shared, *"Our girls get a lot of peer influence and pressure from those who have already undergone FGM/C. This happens against the wish of many parents and especially after the anti-FGM/C seminars have ended. The cut girls harass and abuse the uncut girls whenever they happen to meet. This then makes it difficult for us as parents to convince girls not to undergo FGM/C. We get more pressure from our age-set. . . Then as the parent you are forced to take your young girl for FGM/C before she is mature enough."*

## Shift toward medicalization of FGM/C

The participants' narratives suggest that FGM/C practice has changed in form. Notable shifts were cutting girls at a younger age, less severe cutting and medicalized cutting [10]. Although these shifts are interrelated, in this manuscript, we focus on medicalization, which was

frequently mentioned among the Abagusii and Somali communities but hardly talked about by the Kuria, one of whom noted that medicalization was a "taboo".

According to the participants from the two communities and some of the key informants, FGM/C was increasingly performed by health care providers (doctors, nurse-midwives and clinical officers) at home or in a health facility. The performance of FGM/C at home was more common in rural areas as one nurse from Kisii county explained how parents picked her up to go perform the cutting on their daughters at home *"Both educated and non-educated like teachers, come with a vehicle to pick me to go cut their daughters. Both the rich and poor come for my services"* The performance of FGM/C in private clinic was common in urban areas as supported by sentiments from a clinical officer from Eastleigh, Nairobi. *"Yes, I have performed medicalised FGM in form of sunna. I would like to call it female genital modification or a less severe form or the Sunna (clitoridectomy)".*

Medicalized FGM/C was reportedly driven by beliefs that it reduced the risk of health complications (or ensured that they were addressed in a timely manner), shortened the recovery period, reduced the spread of communicable diseases, such as HIV, was discreet, and enabled women to give birth normally. As one married man who participated in an FGD in Kawangware explained, *"Right now, FGM/C is safe compared to the past because of being performed by doctors. Transmitting of diseases is reduced, and other things like bleeding are less likely because dressing is done to them."* Interview responses from a health care professional from Garissa supported sentiments that medicalization minimized complications associated with FGM/C *"Health complications have reduced for example bleeding, pain and the case of undergoing Caesarean section because of pain relievers and stitching of the cut part".* In justifying medicalization, a clinical officer from Eastleigh reiterated his capacity to perform FGM/C as a medical procedure and to address any occurring complications *"When it is done by medics and it's done under medication the probability of having severe complications is very minimal and in case of any, the probability of solving the complication is very high so there will be a possibility of less complications"* Participants responses suggested that medicalization was more common in urban areas, but that clients who sought medicalized FGM/C came from all socioeconomic backgrounds.

Some participants' comments suggested that medicalization was a response to the law prohibiting FGM/C because girls who underwent medicalized FGM/C healed quickly enabling families to keep the practice hidden. As a 40-year-old woman who was interviewed in Kawangware noted, *"medicalized FGM/C allows families to cut and hide their girls because they do not stay for long while recuperating in the house. For the medicalized FGM/C, no one will even know a girl have been cut."* The fear of the law, may also have informed the location where medicalized FGM/C was performed with participants explaining that girls were either cut in private health facilities or were cut at home by a health care provider. Having a girl cut in a public health facility was considered a risky undertaking. Underscoring this concern, an older woman in an FGD in Eastleigh explained, *"People fear going to public hospital for FGM/C because they may be arrested if it is discovered. No one is ready for public hospitals so most do it in their homes."* However, even health providers from the public sector were noted to engage in the performing FGM/C in the private spaces, underscoring the legal risk involved in performing FGM/C as noted by a nurse from Easleigh. *"FGM/C is done by medical/nurse professionals nowadays, at the private hospital"* Participant responses suggested that the health professionals were aware about the illegality of FGM/C but could easily hide the practice under medical procedures in the justification of preventing a would be major complication associated with FGM/C *"But even then when we were doing FGM/C, we did not care about the illegality of the law because, we knew this is a procedure we were doing at the clinic, we were not*

*advertising it. We were doing it after we've counselled the mothers not to do it and only going ahead to prevent a major risk or a major harm".*

Although some participants reported that medicalized FGM/C boosted health care providers' income and incentivizes them to convince parents to cut their daughters, few health providers disclosed that they performed FGM/C or had done so in the past. In describing the procedure, they noted that they performed "minor" cuts in response to clients' demands and accepted some payments as noted by a Nurse from Kisii "*The health workers are doing FGM/C for money. You are given money by parent and they convince you to cut their girls".* Among those who disclosed that they had performed FGM/C, some explained that they first counselled the client against undergoing FGM/C and only performed the cut if the client insisted. Underscoring these points, a clinical officer working in Eastleigh explained, *"For us it's more of like I said earlier; doing a less severe form of FGM/C. We mainly focused on providing counselling and educating the mothers who were coming to our facility on the effects of severe forms of FGM/ C. If we failed to convince them to abandon FGM/C, we would decide to perform a less severe form of FGM/C."* Some providers were, however, noted to secretly offer medicalized FGM/C because they feared negative sanctions from the community or because they wanted to reduce the negative impacts of the practice, even when they were aware of the illegality of the procedure.

When asked whether medicalization was a step towards abandonment, participants held differing views. Some suggested that medicalization would not lead to the abandonment of the practice and argued that medicalized FGM/C was just 'modernization' of the practice. To them, medicalization would normalize the practice in the long run. The law against FGM/C was also noted to be a driver of medicalized FGM/C, presumably because it could be carried out covertly.

In contrast, the health care providers believed that medicalization of FGM/C would lead to the abandonment of the practice suggesting that medicalization is a transition stage in response to people being more enlightened about the need to abandon FGM/C. They also noted that medicalization also gives health care providers an opportunity to create awareness on the impacts of FGM/C, which they felt would eventually lead to abandonment. However, interview responses from health professionals showed a gap in knowledge on FGM/C issues questioning their capacity to prevent FGM/C, sentiments supported by a Nurse from Kisii "*FGM/C was just mentioned under child abuse; when I did my certificate and my diploma Nursing courses"* Similar gaps were raised about the clinical officer training program in supporting the view *"I am not sure that there's any training on FGM prevention strategies am not aware who's doing it or where it's done".* Although there was knowledge gap possibly emanating from inadequate content of FGM/C components in the training curricula, there was motivation not to practice medicalisation informed by moral-ethical principles as noted by sentiments from a male Nurse from Kuria *"The reason is that FGM is a criminal and unethical according to our professional".* Indeed, a Kisii participant responses show that professional bodies can deter medicalisation *"I decided I don't want to continue as I heard there is penalty and a fine for practicing FGM/C. Your practice license can be withdrawn if implicated in medicalised FGM/C".*

Our study findings provide an opportunity to better understand FGM/C and the changes related to the practice. The findings highlight medicalization of FGM/C—a change in nature of FGM/C possibly triggered by the legal sanctions against the practice as well as a publicized narrative on the health risks associated with the practice. Although medicalization is not a new phenomenon, recently it received attention because of children's right violation, impact on abandonment efforts, and the fact that it is perpetuated by professionals who should be championing the rights of the vulnerable. We show that families seek FGM/C for their daughters because of reasons that converge across ethnic groups—conformity to culture and tradition,

marriageability and ensuring chastity—or that differ depending on ethnic group (e.g., religion or as a rite of passage). Consistent with previous evidence suggesting that social pressure and tradition are the most compelling factors for continuation of FGM/C, [7, 29, 30] we found that FGM/C among the Abagusii, Kuria and Somali communities in Kenya is largely driven by social norms, which are enforced through social pressure and negative sanctions for non-compliance. [31] Within these communities, FGM/C is considered meaningful as those who undergo the practice conform to cultural expectations and receive social significance, a sense of identity, and respectability as an ideal member of the community. [32] Although there is convergence of the norms, there are differentials in the context of the FGM/C practice. Whereas among the Abagusii and Somalis FGM/C is performed on young girls, privately, at home and mainly by health care providers, the Kuria girls are cut relatively older, in public and in groups of girls under community decision made under the discretion of council of elders. These findings present opportunities for interventions through targeted strategies that focus on health care providers, family decision makers, religious/opinion leaders and male involvement in Abagusii and Somali communities. Whereas in the Kuria community strategies targeting the council of elders, alternative rite of passage for the girls as well as male involvement to change their altitude towards marriage of uncut women could have a greater impact in FGM/C abandonment.

The findings demonstrate adoption of medicalization among the Abagusii and Somali communities, which is notably driven by concerns about FGM/C-related complications and the legal banning of the practice. Similar reasons have been documented in the literature. [7, 9, 18] Medicalization was commonly reported in urban compared to the rural settings. This could be attributed to urbanization, modernization, education and migration as well as inter community fusion. This is consistent with findings that the educated and urban families prefer less cutting for their daughters (Islam & Uddin, 2001). As a result of the aforementioned factors, there could be shift in social norms prompting a change in the practice of FGM/C. As regard the Kuria community, only the rural setting was studied with no medicalization noted. Whether the community is steps behind before it adopts medicalization is subject of another research. While proponents of medicalized FGM/C submit that the practice prevents complications, healing is faster and enhance good parental control on the girl, it is not entirely true. Findings by Bjälkander and colleagues found that girls who undergo FGM/C before 10 years of age seem to be more vulnerable to serious complications than those who are older at the time of cutting. [29] Indeed, the excised fresh is likely to be more extensive during medicalization due to the fact that the girl is under anesthesia thus the sensation/resistance to pain is moderated. Thus interventions adopting development approach for communities as well capacity building for health care providers with emphasis to the do no harm principle should be scaled up.

The motivations for health care providers to perform medicalized FGM/C were consistent with documented evidence in other studies—financial gain, [13, 14] reduction of health complications [1, 13, 18, 33, 34], and clients' demands for medicalized FGM/C. [35, 36] Indeed, while responses showed health care providers were motivated by financial gain among others, the latter two factors appears to play a central role. First the performer of medicalized FGM/C hailed from the same community as the girl. Second, the cost for medicalized FGM/C was way below what was charged by traditional cutters. Thus, health professional did not simply want extra income, but were worried that clients would turn to someone not qualified to perform the cutting, or may end up being exposed to more severe cutting. This proposal could fit the concept of harm reduction [18]–that is, promoting no cutting at all, but if the client refuses, offering the least harmful acceptable option. The concept is consistent with evidence from Somaliland that suggested that a shift to nicking was better than infibulation for clients who

insisted on having FGM/C performed.[37] However, the concept negates and is contrary to the unequivocal stand of zero tolerance policy on medicalization by WHO.[7] Thus strategies for addressing medicalization should consider the impacts of policy decision on the FGM/C practice as well as be context specific. Although the narratives suggest that medicalization of FGM/C is increasing, the preferred venue was the girls' home and, occasionally, private health facilities. Performing FGM/C at home or health clinic enabled families to maintain secrecy while concealing the activities of the health providers because of illegal nature of FGM/C. Taken together the findings suggests there is increasing adoption of medicalization as well as clinicalization (performance of FGM/C by health provider in health clinics) [38] among the Abagusii and Somali communities. Indeed, our findings showed some clinicians who performed FGM/C in the guess of female genital modifications. This finding have been elicited elsewhere in Egypt where some doctors debated whether FGM/C was medically necessary and were using the term female genital cosmetic surgery.[39] Similarly, some clinician can hide behind the laws that legalize plastic surgery in Kenya to advance medicalized FGM/C as well as capitalize on the part of the definition for FGM/C under the prohibition of FGM Act namely "but does not include a sexual reassignment procedure or a medical procedure that has a genuine therapeutic purpose".[40] However, for this to happen medicalization must present a financially lucrative opportunity that may only be sustained by economically capable adult women as opposed to majority of girls who cannot afford the cost of surgeons. Taken together, these findings suggest that medicalization enables families who may be questioning the value of FGM/C to continue to conform to social norms, but in a way that protects girls from health risks and the family from legal risks.

Efforts to curb medicalization in contexts where it is predominately carried out at home may face considerable challenges. These challenges might explain why in 1994 Egyptian authorities permitted FGM/C to be performed on girls in designated facilities at fixed times and prices to mitigate complications and eventually end the practice [18, 41], the so called clinicalization. However, subsequent pressure from international agencies, as well as the reported deaths of girls who were cut in hospitals, instigated a renewed ban on the medicalized FGM/C in public hospitals. [1] These findings call for all-inclusive consultation, taking into consideration existing policies and best practices to facilitate the issuance of well-reasoned guidelines. Additionally, since medicalization was performed mainly in private health clinics, this offers the health system an opportunity to act through the introduction of a system to dialogue, capacity building, monitoring, reporting, and tracking FGM/C activities in all health facilities.

Successful abandonment of FGM/C and related medicalization requires health care champions that are fully equipped to address the prevention and management of the practice. The knowledge, skills, practices and attitudes should be inculcated during professional training or through in-service trainings. Our findings found that FGM/C components were not adequately addressed in the health care professional training, while their training curricula was inadequate in covering strategies to address FGM/C. The findings mirror gaps in knowledge levels on FGM/C components among Nurse-Midwives drawn from high prevalent counties in Kenya.[9] The knowledge level was similar to those of Midwives from Sudan that associated their challenges with only a brief mention of female circumcision during their professional training.[42] However, the challenges in knowledge may be corrected by on job training through encountering and practical management of FGM/C cases [42, 43]- an indication of the importance of in service training. However, this need to tailored to suit the workload and service disruptions in already understaffed health facilities. Thus innovative strategies that address this challenges can have high success rates.

There were divergent views regarding the role of medicalization in abandonment of FGM/C. Some viewed medicalization as modernization of the practice and believed it would

normalize the practice thus making it difficult to end FGM/C. Some scholars have argued that medicalization creates a tacit approval for FGM/C thus promoting its continuation and making the process of complete abandonment more difficult. [11, 44] Others note that it creates the impression that FGM/C can be performed safely and is condoned by respected health care professionals, thus reducing motivation of families to abandon the practice. [7, 13, 34] On the other hand, some participants noted that the shifts in the practice would lead to FGM/C abandonment because they observed them as representing transitional steps. This is consistent with reports that have speculated that medicalization or less severe cutting can be an interim step toward abandonment. [37] Accordingly, medicalization is seen to present an opportunity for health care providers to create awareness on the impacts of FGM/C in line with the desire to reduce the chance of complications and drawing attention to the practice [14], which can eventually lead to the total eradication of the practice.

The divergent views on the role of medicalization in the abandonment of FGM/C shifts focus on the health sector to take a strategic decision to lead from the front in confronting this practice that has far reaching ramifications in the campaign against FGM/C. The findings present opportunities for leveraging namely; mainstream FGM/C components in the medical and Nursing training curricula as well continuous professional development programs, structured professional dialogues and discourses to address social, professional and legal norms underpinning medicalized FGM/C; Collaboration and networking with strategic sectors such as legal, law enforcement, education, media, professional regulatory bodies and internal security to address medicalized FGM/C; and Sensitization and awareness creation on the harms of FGM/C and medicalization.

Although this was a qualitative study from four study sites involving only three ethnic groups and the findings may not be generalizable, the design was robust enough enabling results that presents new insights into Medicalization. Indeed, the study findings bring to the fore concerns about the medicalization of FGM/C, which may pose challenges to efforts promoting the abandonment of the practice. However, the findings should be interpreted in light of some limitations because FGM/C is illegal in Kenya and thus some participants may have been unwilling to openly share their views about the practice because they feared legal and professional repercussions.

## Conclusions

The practice of FGM/C is pervasive in the studied Kenyan communities albeit substantial medicalization. Although the practice and its related changes have been perpetuated by generational linked social norms, negative sanctions, religion, and marriageability, there are emerging factors in response to legal sanctions and growing awareness of the health risks prompting families and health professionals to perpetuate FGM/C. This underscores the need to tailor FGM/C interventions with consideration for dynamics and specific context in abandonment efforts in FGM/C practice and its related shifts. The strategies should target law enforcement, health care providers, custodians of culture in the community, religious leaders, males as well as women/girls in FGM/C prevention programs.

## Supporting information

**S1 File. FGD with women from Garissa.**
(PDF)

**S2 File. KII with government officer from Kuria.**
(PDF)

**S3 File. IDI with a guardian from Kisii.**
(PDF)

**S4 File. IDI with a father of a girl from Kawangware.**
(PDF)

**S5 File. KII with a health provider from Kawangware.**
(PDF)

## Acknowledgments

The authors thank the following for their invaluable contribution: Jerry Okal for his support during the qualitative data analysis and Bettina Shell-Duncan for her critical review of the work.

## Author Contributions

**Conceptualization:** Samuel Kimani, Jacinta Muteshi, Jaldesa Guyo.

**Formal analysis:** Samuel Kimani.

**Funding acquisition:** Jacinta Muteshi.

**Investigation:** Samuel Kimani.

**Methodology:** Samuel Kimani, Caroline W. Kabiru, Jacinta Muteshi.

**Supervision:** Samuel Kimani.

**Validation:** Samuel Kimani, Jaldesa Guyo.

**Writing – original draft:** Samuel Kimani, Caroline W. Kabiru.

**Writing – review & editing:** Samuel Kimani, Caroline W. Kabiru, Jacinta Muteshi, Jaldesa Guyo.

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
