## [Decision Letter · Decision Letter 0]

13 Nov 2019

PONE-D-19-27940

Female Genital Mutilation/Cutting: Emerging factors sustaining medicalization related changes in selected Kenyan communities

PLOS ONE

Dear Dr Kimani,

Thank you for submitting your manuscript to PLOS ONE. After careful consideration, we feel that it has merit but does not fully meet PLOS ONE’s publication criteria as it currently stands. Therefore, we invite you to submit a revised version of the manuscript that addresses the points raised during the review process.

Please see comments from reviewers for changes that are needed prior to acceptance. We look forward to reading your updated manuscript.

We would appreciate receiving your revised manuscript by Dec 28 2019 11:59PM. To enhance the reproducibility of your results, we recommend that if applicable you deposit your laboratory protocols in protocols.io, where a protocol can be assigned its own identifier (DOI) such that it can be cited independently in the future. For instructions see: http://journals.plos.org/plosone/s/submission-guidelines#loc-laboratory-protocols

We look forward to receiving your revised manuscript.

Kind regards,

Amy Michelle DeBaets, PhD

Academic Editor

PLOS ONE

Journal Requirements:

2. We note that you have indicated that data from this study are available upon request. PLOS only allows data to be available upon request if there are legal or ethical restrictions on sharing data publicly. For information on unacceptable data access restrictions, please see http://journals.plos.org/plosone/s/data-availability#loc-unacceptable-data-access-restrictions

b) If there are no restrictions, please upload the minimal anonymized data set necessary to replicate your study findings as either Supporting Information files or to a stable, public repository and provide us with the relevant URLs, DOIs, or accession numbers. Please see http://www.bmj.com/content/340/bmj.c181.long for guidelines on how to de-identify and prepare clinical data for publication. For a list of acceptable repositories, please see http://journals.plos.org/plosone/s/data-availability#loc-recommended-repositories

3. We noted in your submission details that a portion of your manuscript may have been presented or published elsewhere. [The data are from a larger cross-sectional qualitative study that sought to understand shifts in the practice of FGM/C in selected practicing communities in Kenya. The publication can be found in www.popcouncil.org]

Please clarify whether this publication was peer-reviewed and formally published. If this work was previously peer-reviewed and published, in the cover letter please provide the reason that this work does not constitute dual publication and should be included in the current manuscript.

Reviewers' comments:

Reviewer's Responses to Questions

**Comments to the Author**

1. Is the manuscript technically sound, and do the data support the conclusions?

Reviewer #1: Yes

Reviewer #2: Yes

2. Has the statistical analysis been performed appropriately and rigorously? 

Reviewer #1: N/A

Reviewer #2: N/A

3. Have the authors made all data underlying the findings in their manuscript fully available?

Reviewer #1: Yes

Reviewer #2: No

4. Is the manuscript presented in an intelligible fashion and written in standard English?

Reviewer #1: Yes

Reviewer #2: Yes

5. Review Comments to the Author

Reviewer #1: This is an important manuscript, covering an issue that requires urgent attention. However, it can be improved if the following issues are addressed:

1. The sentence in the abstract, which starts in line number 21 as..."The transformed FGM/C...." and ends on page 22 must be revised. Its meaning is vague or hanging.

2. The authors should consider including a chart under study participants section. This chart should depict a distribution of participants by some dominant characteristic(s)e.g. age, ethnicity, etc...as deemed by the authors.

3. Line number 188 has some mention of demographic data; which demographic variables are these? There is need to mention this

4. The conclusion is weak as presented: it is too summarised and needs to be expanded. For example, the factors that have been passed from one generation to another should be highlighted. Examples of effective interventions could be provided.

Reviewer #2: General comments

This study assessed drivers of the medicalization of FGM. In general, this is a well-written manuscript. However, I am not convinced that the medicalization of FGM is new or that it is a recent change to the practice of FGM as portrayed by the authors. Also, the results fail to provide adequate background to the practice of FGM in the ethnics groups studied. Is FGM done at birth or as a right of passage during puberty? If it is done at birth, what role does maintain culture play and what influence do mothers-in-law or grandmothers play in maintaining the culture. If done as a right of passage, at what age is it done? Are there rituals performed? what are the significance of the ceremony?

An underdeveloped piece in this paper is the role of medical personnel's in performing FGM. If the practice is illegal and it is against medical guidelines, why do health providers continue this practice? How does this impede or enhance efforts to end FGM? Are the health providers in favour of FGM? Were they instructed to end the practice? How much do they make from performing FGM? How does the money they make present additional barriers to ending FGM?

Even though the study included key informants, the analysis did not adequately delineate the views of key informants, especially doctors and nurses. Even though it was mentioned that they try to convince women on the adverse effects of FGM, it is unclear if they discuss the legal implications with women. I would like to see a more nuanced presentation and interpretation of results that not only focus or emphasized the perspective of women but also that of the key informants.

The discussion of findings could improve a great deal. Given the study findings, what implications could be drawn to help eliminate FGM practice? The study results are not new, but the method is robust enough to present the results in a way that make new analysis possible. For instance, authors could discuss what new insights they learn from their results. Also, the authors could focus more on the relevance of their results in developing models to end the practice of FGM.

Introduction

There is a need to presents a review of drivers of the medicalization of FGM in previous studies and settings in SSA.

Methods

There is a need to follow Consolidated criteria for reporting qualitative research (COREQ).

What questions were asked of participants including doctors and nurses?

Results

There is a need to present more results on the perspective of key informants such as doctors and nurses. Authors mention deductive and inductive analysis but I did not see evidence of this in the results.

Discussion

The discussion could improve by focusing on new insights drawn from the study results, the implications of the results regarding FGM and what models to recommend to support the fight to end FGM. Even though this study is a qualitative study, but the study included some Somalian, authors should comment on the relevance of the results in understanding the FGM situation on other SSA settings.

Authors should not only emphasized the limitation of the study but also discuss the strength of the study. The diverse groups of participants, for me, is a strength of this study.

I hope these comments and questions would help the authors to present a more nuanced analysis in this paper.

6. PLOS authors have the option to publish the peer review history of their article (what does this mean?). If published, this will include your full peer review and any attached files.

**Do you want your identity to be public for this peer review?** For information about this choice, including consent withdrawal, please see our Privacy Policy

Reviewer #1: No

Reviewer #2: Yes: Anthony Ajayi

While revising your submission, please upload your figure files to the Preflight Analysis and Conversion Engine (PACE) digital diagnostic tool, https://pacev2.apexcovantage.com/ PACE helps ensure that figures meet PLOS requirements. To use PACE, you must first register as a user. Registration is free. Then, login and navigate to the UPLOAD tab, where you will find detailed instructions on how to use the tool. If you encounter any issues or have any questions when using PACE, please email us at figures@plos.org. Please note that Supporting Information files do not need this step.

---

## [Author Response · Author response to Decision Letter 0]

28 Dec 2019

MEMORANDUM

28th December, 2019

To: Editor, PLOS ONE Journal

From: Dr. Samuel Kimani, Corresponding author

cc: Dr. Caroline Kabiru, Jacinta Muteshi and Prof Jaldesa Guyo

Re: Summary of changes made to manuscript Number: PONE-D-19-27940

Thank you again for your comments and the opportunity to resubmit a revised draft of

our manuscript PONE-D-19-27940 titled “Female Genital Mutilation/Cutting: Emerging factors sustaining medicalization related changes in selected Kenyan communities”

We appreciate the interest and time that you and the reviewers have dedicated in our manuscript and the constructive criticism they have given. The insightful comments have been very helpful and are all addressed in our revision. We have highlighted the changes within the manuscript in YELLOW while an unmarked copy have also been submitted. Kindly find below our point-by-point responses to the editor and reviewers’ concerns.

Thank you again for consideration of our revised manuscript and we hope that it will be suitable for PLOS ONE Journal. We look forward to hearing from you regarding our submission and to respond to any further questions and comments you may have.

With this cover letter, the revised manuscript Number: PONE-D-19-27940 “Female Genital Mutilation/Cutting: Emerging factors sustaining medicalization related changes in selected Kenyan communities” is enclosed. 

All the comments have been addressed in the relevant pages as indicated in the response below.

Actions Required followed by Responses 

Comments from the editor 

1.Please ensure that your manuscript meets PLOS ONE's style requirements, including those for file naming. This has been addressed throughout the manuscript 

2. We note that you have indicated that data from this study are available upon request. PLOS only allows data to be available upon request if there are legal or ethical restrictions on sharing data publicly. * In your revised cover letter, please address the following prompts:

b) If there are no restrictions, please upload the minimal anonymized data set necessary to replicate your study findings as either Supporting Information files or to a stable, public repository and provide us with the relevant URLs, DOIs, or accession numbers. Please see http://www.bmj.com/content/340/bmj.c181.long for guidelines on how to de-identify and prepare clinical data for publication. For a list of acceptable repositories, please see http://journals.plos.org/plosone/s/data-availability#loc-recommended-repositories

There is no restriction to the availability of data, There is no ethical or legal restrictions, Anonymized data set have been uploaded

3. We noted in your submission details that a portion of your manuscript may have been presented or published elsewhere. [The data are from a larger cross-sectional qualitative study that sought to understand shifts in the practice of FGM/C in selected practicing communities in Kenya. The publication can be found in www.popcouncil.org]

Please clarify whether this publication was peer-reviewed and formally published. If this work was previously peer-reviewed and published, in the cover letter please provide the reason that this work does not constitute dual publication and should be included in the current manuscript. The published report is a 50 pager abridged version of a comprehensive 260 pages’ report. 

The report was internally and externally reviewed as a requirement by the funder for clarity and to meet the editorial requirement. 

The review is not necessarily rigorous as that of a journal publication while the reviewers could be both persons we have interacted before.

The publication does not constitute dual publication 

 Comments from Reviewer #1: 

1. The sentence in the abstract, which starts in line number 21 as..."The transformed FGM/C...." and ends on page 22 must be revised. Its meaning is vague or hanging.This has been addressed in the abstract in lines 21-22 as follows: Transformation of FGM/C include medicalization although poorly understood has increased undermining abandonment efforts for the practice.

2. The authors should consider including a chart under study participants section. This chart should depict a distribution of participants by some dominant characteristic(s)e.g. age, ethnicity, etc...as deemed by the authors. This has been addressed in the Materials and methods section in pages 11-12 lines 209-210

3. Line number 188 has some mention of demographic data; which demographic variables are these? There is need to mention this We provide more information on demographic variables in line 224-226 as follows: Demographic characteristics data namely gender, age, ethnicity, marital status, location as well as cadre, work station and organization represented for the participants were entered in anonymized form into password-protected Excel spreadsheets and descriptively analyzed. 

4. The conclusion is weak as presented: it is too summarised and needs to be expanded. For example, the factors that have been passed from one generation to another should be highlighted. Examples of effective interventions could be provided.

We strengthen the conclusion in pages 27-28, lines 570-578 as follows: The practice of FGM/C is pervasive the studied Kenyan communities albeit substantial medicalization. Although the practice and its related changes have been perpetuated by generational linked social norms, negative sanctions, religion, and marriageability, there are emerging factors in response to legal sanctions and growing awareness of the health risks prompting families and health professionals to perpetuate FGM/C. This underscores the need to tailor FGM/C interventions with consideration for dynamics and specific context in FGM/C and related shifts abandonment efforts. The strategies should target law enforcement, health care providers, custodians of culture in the community, religious leaders, males as well as women/girls in FGM/C prevention programs.

Comments from Reviewer #2: 

5. This study assessed drivers of the medicalization of FGM. In general, this is a well-written manuscript. However, I am not convinced that the medicalization of FGM is new or that it is a recent change to the practice of FGM as portrayed by the authors. This has been revised throughout the manuscript with a specific mention on page 21 lines 424-429: The findings highlight medicalization of FGM/C - a change in nature of FGM/C possibly triggered by the legal sanctions against the practice as well as a publicized narrative on the health risks associated with the practice. Although medicalization is not a new phenomenon, recently it received attention because of children’s right violation, impact on abandonment efforts, and the fact that it is perpetuated by professionals who should be championing the rights of the vulnerable.

6. Also, the results fail to provide adequate background to the practice of FGM in the ethnics groups studied. Is FGM done at birth or as a right of passage during puberty? If it is done at birth, what role does maintain culture play and what influence do mothers-in-law or grandmothers play in maintaining the culture. If done as a right of passage, at what age is it done? Are there rituals performed? what are the significance of the ceremony? Although some details were already there, we have strengthened and provided more information on the background of ethnic groups in Methods section pages 8-9 lines 149-176 as follows:

7. An underdeveloped piece in this paper is the role of medical personnel's in performing FGM. If the practice is illegal and it is against medical guidelines, why do health providers continue this practice? How does this impede or enhance efforts to end FGM? Are the health providers in favour of FGM? Were they instructed to end the practice? How much do they make from performing FGM? How does the money they make present additional barriers to ending FGM? This has been addressed in the discussion section

8. Even though the study included key informants, the analysis did not adequately delineate the views of key informants, especially doctors and nurses. Even though it was mentioned that they try to convince women on the adverse effects of FGM, it is unclear if they discuss the legal implications with women. I would like to see a more nuanced presentation and interpretation of results that not only focus or emphasized the perspective of women but also that of the key informants. We have presented responses on medicalization from Key informants especially the health care providers in diverse paragraphs as follows:

Pages 17 lines 335-342

Pages 18 lines 349-357

Pages 19 lines 372-381

Pages 19 lines 385-391

Pages 20 lines 392-398

Pages 20 lines 408-414

Pages 21 lines 415-422

9. The discussion of findings could improve a great deal. Given the study findings, what implications could be drawn to help eliminate FGM practice? The study results are not new, but the method is robust enough to present the results in a way that make new analysis possible. For instance, authors could discuss what new insights they learn from their results. Also, the authors could focus more on the relevance of their results in developing models to end the practice of FGM. We have strengthened the discussion as per the comments as follows:

Pages 22 lines 439-448: Although there is convergence of the norms, there are differentials in the context of the FGM/C practice. Whereas among the Abagusii and Somalis FGM/C is performed on young girls, privately, at home and mainly by health care providers, the Kuria girls are cut relatively older, in public and in groups of girls under community decision made under the discretion of council of elders. These findings present opportunities for interventions through targeted strategy that focus on health care providers, family decision makers, religious/opinion leaders and male involvement in Abagusii and Somali communities. Whereas in the Kuria community strategies targeting the council of elders, alternative rite of passage for the girls as well as male involvement to change their altitude towards marriage of uncut women could have a greater impact in FGM/C abandonment. Other changes are found in the following parts of the manuscript. 

Pages 22 lines 453-446

Pages 23 lines 461-468

Pages 23 lines 473-483

Pages 24 lines 484-501

Pages 25 lines 519-528

Pages 26 lines 529-532

10. Introduction

There is a need to presents a review of drivers of the medicalization of FGM in previous studies and settings in SSA. We addressed this in page 5 lines 86-95 as follows: Emerging evidence from demographic and health surveys (DHS), multiple indicator cluster surveys (MICS), and qualitative research suggests that some families and communities are shifting how FGM/C is practiced, to sustain rather than abandon it, mainly due to – reduce the health risks, willingness of some health providers to carry out the procedure, financial incentive or social recognition. [6, 7, 9] The greatest burden of medicalized FGM/C is concentrated in Sudan (67%), Egypt (38%), Guinea (15%), Kenya (15%) and Nigeria (13%) where nurses, trained midwives, or other lower-level providers perform FGM/C. [6, 9] Furthermore, the risk of medicalization is higher among girls aged 0-14 years than among women aged 15-49 years and a trend towards medicalization is more likely to institutionalize the practice and encourage its continuation rather than abandonment. [7, 12, 13]

11. Methods

There is a need to follow Consolidated criteria for reporting qualitative research (COREQ).

What questions were asked of participants including doctors and nurses? We have addressed COREQ throughout the manuscript to the best of our ability

12. Results

There is a need to present more results on the perspective of key informants such as doctors and nurses. Authors mention deductive and inductive analysis but I did not see evidence of this in the results. We have presented responses on medicalization from Key informants especially the health care providers in diverse paragraphs as follows:

Pages 17 lines 335-342

Pages 18 lines 349-357

Pages 19 lines 372-381

Pages 19 lines 385-391

Pages 20 lines 392-398

Pages 20 lines 408-414

Pages 21 lines 415-422

13. Discussion

The discussion could improve by focusing on new insights drawn from the study results, the implications of the results regarding FGM and what models to recommend to support the fight to end FGM. Even though this study is a qualitative study, but the study included some Somalian, authors should comment on the relevance of the results in understanding the FGM situation on other SSA settings. This has been addressed in the discussion section lines 473-501

14. Authors should not only emphasized the limitation of the study but also discuss the strength of the study. The diverse groups of participants, for me, is a strength of this study. We addressed this in page 27 lines 562-564 to read as follows: Although this was a qualitative study from four study sites involving only three ethnic groups and the findings may not be generalizable, the design was robust enough enabling results that presents new insights into Medicalization. Indeed, the study findings bring to the fore concerns about the medicalization of FGM/C, which may pose challenges to efforts to promote the abandonment of the practice.

15. A thorough re-look into the manuscript including English proof reading has been carried out.

Sincerely,

Dr. Samuel Kimani

---

## [Editor Report · Decision Letter 1]

15 Jan 2020

Female Genital Mutilation/Cutting: Emerging factors sustaining medicalization related changes in selected Kenyan communities

PONE-D-19-27940R1

Dear Dr. Kimani,

We are pleased to inform you that your manuscript has been judged scientifically suitable for publication and will be formally accepted for publication once it complies with all outstanding technical requirements.

With kind regards,

Amy Michelle DeBaets, PhD

Academic Editor

PLOS ONE

Additional Editor Comments (optional):

Thank you for submitting your revised article to PLOS ONE. We are pleased to see the requested revisions have been made and are happy to accept the article for publication.
---

## [Editor Report · Acceptance letter]

18 Feb 2020

PONE-D-19-27940R1 

Female Genital Mutilation/Cutting: Emerging factors sustaining medicalization related changes in selected Kenyan communities 

Dear Dr. Kimani:

I am pleased to inform you that your manuscript has been deemed suitable for publication in PLOS ONE. Congratulations! Your manuscript is now with our production department. 

With kind regards,

on behalf of

Dr. Amy Michelle DeBaets 

Academic Editor

PLOS ONE